# Taxonomic and Functional Diversity of Flower-Visiting Insects in Coffee Crops

**DOI:** 10.3390/insects15030143

**Published:** 2024-02-21

**Authors:** Juan Diego Maldonado-Cepeda, Jesús Hernando Gómez, Pablo Benavides, Juliana Jaramillo, Zulma Nancy Gil

**Affiliations:** 1Department of Entomology, National Coffee Research Center—Cenicafé, Manizales 170009, Colombia; juandiegomaldonadoc@gmail.com (J.D.M.-C.); jhgomezl@unal.edu.co (J.H.G.); pablo.benavides@cafedecolombia.com (P.B.); 2Theme Lead Regenerative Agriculture, Rainforest Alliance, De Ruyterkade 6, 1013 BG Amsterdam, The Netherlands; jjaramillo@ra.org

**Keywords:** bees, biodiversity, *Coffea arabica*, flower visitor, functional diversity, pollinators

## Abstract

**Simple Summary:**

Floral-visiting insects and pollinators play an important role in ecosystem services, and consequently, their identification and study are essential to their adequate preservation and management in crops of agricultural interest. Coffee is a worldwide commodity; however, the diversity of insects that visit its flowers has been little studied. The objective of this research was to quantify the abundance, richness, and functional diversity of coffee floral-visiting insects, especially bees. The results showed that coffee crops host a wide diversity of flower visitors, especially bees, which could be beneficial for productivity and contribute to the maintenance of plant species that accompany coffee cultivation.

**Abstract:**

Despite the important role that flower-visiting insects play in agricultural production, none of the previous studies of coffee pollinators in Colombia have incorporated functional diversity into their analysis. Therefore, this study aimed to quantify the abundance, richness, and functional diversity of insects that visit flowers in coffee crops. Twenty-eight plots were selected among five sites in the north, center, and south of Colombia. In each plot, coffee flower insect visitors were collected and recorded on 90 trees at eight-minute intervals per tree, at three different times over three days. All sampling was carried out during two flowering events per year, over three years, resulting in a total of 1240 h of observations. Subsequently, the insects were taxonomically identified, and the number of individuals and species, as well as the diversity of the order q, were estimated. Functional diversity was also characterized in the bee community. The results: (a) 23,735 individuals belonging to 566 species were recorded; of them, 90 were bees, with the native species being the most abundant during 10:30 and 13:00 h; (b) bees formed five functional groups, with corbiculate and long-tongued non-corbiculate bees being the most abundant and occupying the largest regions of functional space; (c) potential pollinators in coffee crops are *Apis mellifera*, *Nannotrigona gaboi*, *Tetragonisca angustula*, *Geotrigona* cf. *tellurica*, and *Partamona* cf. *peckolti*. Coffee crops host a wide diversity of flower visitors, especially bees, which could be beneficial for productivity and contribute to the maintenance of plant species that accompany coffee cultivation.

## 1. Introduction

Insects form an important part of ecosystems as they provide different services, particularly pollination. According to Ollerton et al. [1], 87.5% of angiosperms are pollinated by animal vectors, most of which are insects. 

For this reason, it is essential to study the taxonomic and functional diversity of flower-visiting insects. Because insect richness and density are the most important crop yield predictors [2,3], it is fundamental to study the taxonomic diversity of flower-visiting insects [2,3]. According to Klein et al. [4], there are three ways to use this information to increase crop yield: (1) random inclusion of the most efficient and effective species in pollination; (2) complementary pollination among species; and (3) facilitation, as several species could facilitate the permanence or role of others as pollinators of a given crop, and a better understanding of the taxonomic diversity of flower visitors can serve to successfully adjust the estimates of insects present in a given region or crop. 

Additionally, functional diversity provides a valuable understanding of the role of species and allows the incorporation of characteristics that influence their performance and processes in the ecosystem [5]. In addition, it allows us to know the response of species to climate change [6]. Coffee is an important agricultural commodity, produced in about 80 tropical countries, with an estimated 125 million people depending on it for their livelihoods. In Colombia, coffee (*Coffea arabica* L., *Rubiaceae*) is of particular interest to researchers, mainly because of its contribution to the national economy and because it is the livelihood of almost half a million coffee-growing families nationwide [7]. Coffee, a self-pollinated plant, grows best between 1200 and 1800 m above sea level and at temperatures between 18 and 21 °C [8]. Flowers open in the morning hours and can last one or two days after being pollinated. Non-pollinated flowers can last up to five days [9]. 

Despite the autogamy of *C. arabica*, several groups of flower visitors are known to visit coffee crops [2]. Studies to identify the main flower visitors of coffee crops carried out in Brazil, Costa Rica, Ecuador, Indonesia, and Mexico revealed that social bees belonging to the tribes Meliponini and Apini were the most important in the pollination of *C. arabica*, in particular those of the genus *Apis* [9]. Moreaux et al. [10] also performed a global analysis of the value of biotic pollination in *C. arabica*. In the studies they reviewed, bees were found to represent an average of 70% of all recorded species and 87% of all individuals. Based on these results, bees are believed to be the main pollinators of coffee [10]. 

Jaramillo [11] evaluated the diversity and abundance of bees in *C. arabica* crops in the department of Antioquia, Colombia, and found three (*Apidae*, *Halictidae*, and *Megachilidae*) of the five families reported for the country, distributed in twenty genera and fifty species. Of these, 45% were solitary bees and 19% were stingless social bees from the tribe Meliponini, with the family *Apidae* being the most abundant. Subsequently, a second study by Cepeda et al. [12] evaluated the diversity of coffee-visiting bees in the department of Cundinamarca, Colombia. They found thirteen species of the same three families found by [11]. The family with the highest richness was *Apidae*, with nine species, particularly in the genus *Trigona*. The species representing the highest number of visits to coffee were *Apis mellifera* and *Paratrigona eutaeniata* Camargo & Moure, 1994 (*Hymenoptera*: *Apidae*) [12]. Although bees are the most abundant flower visitors in coffee, the literature reports other groups of insects, for example, species of Syrphidae (*Diptera*) [13,14], *Hesperiidae*, and *Nymphalidae* (*Lepidoptera*), Dictyoptera, and *Coccinellidae* (*Coleoptera*) [13,15], and several wasps of the *Hymenoptera*.

There are no known studies on functional diversity in coffee; however, several studies have been carried out on other crops. Garibaldi et al. [16] found that bee species with functional traits adjusted to the requirements of flowers were the best pollinators in 33 crops evaluated. It is also important to mention the meta-analysis carried out by Woodcock et al. [17] in canola *Brassica napus* L. (*Brassicaceae*), where the complementary pollination among species was corroborated, indicating a community presenting non-overlapping traits favors the pollination service. Considering that only a few studies have been conducted on insect diversity in *C. arabica* flowers and that it is important to identify which insects visit coffee flowers and the role they play within the crop, the present study focused on estimating and characterizing the taxonomic and functional diversity of flower-visiting insects in coffee crops.

## 2. Materials and Methods

### 2.1. Study Sites

This study was carried out at five sites corresponding to experiment stations of Colombia’s National Coffee Research Center (Cenicafé, its Spanish acronym), located in northern (Cesar and Santander departments), central (Caldas and Quindío departments), and southern (Cauca department) Colombia (Table 1, Figure 1).

### 2.2. Methodology

#### 2.2.1. Taxonomic Diversity

Twenty-eight coffee plots of *Coffea arabica* var. Castillo^®^ were selected in five locations in northern, central, and southern Colombia (Table 1). They were sampled between 2019 and 2022 as follows: two at Pueblo Bello Experiment Station, four at San Antonio Experiment Station (northern), eight at Paraguaicito Experiment Station, ten at Naranjal Experiment Station (central), and four at Manuel Mejía Experiment Station (southern). All coffee plots were grown in places exposed to the sun, except at the Pueblo Bello Experimental Station, where coffee was grown in the shade. In Colombia, coffee plots are surrounded by areas of natural vegetation, patches of secondary forest, and other crops (for example, see Figure 2).

In each coffee plot, ninety coffee plants were randomly selected, and for three consecutive days, all flower-visiting insects were collected from thirty plants per day of flowering. During the samplings, flower-visiting insects were observed and recorded at three different times (7:00 to 9:30, 10:30 to 13:00, and 14:00 to 16:30 h) at 8-min intervals for each plant, totaling 1240 h of observation. No captures were made for *Apis mellifera* L. (*Hymenoptera*: *Apidae*), given its abundance and ease of identification directly from the field. 

The collected insects were placed and tagged in glass vials. Bees were dry-preserved to analyze pollen load, but other taxonomic groups were preserved in 76% ethanol. All specimens were transferred to Cenicafe’s Entomology Laboratory and identified under a Nikon SMZ 1500 stereomicroscope (Nikon Instruments Inc., Melville, NY 11747-3064, USA), following the taxonomic keys proposed by [19,20,21,22]. With the registered information, for each study site and sampling time, the number of species and the number of individuals per species were determined.

#### 2.2.2. Functional Diversity

The functional diversity of the bee community was characterized. To do so, soft effector functional traits, focused on the pollination of *C. arabica*, were determined for each species. Several traits were adapted from those already proposed by [17,23,24], while others were obtained from the information recorded in this study. Table 2 describes the selected traits, which are nesting habit, degree of sociability, size, stigmal contact, location of pollen load structure, type of tongue, abundance from 7:00 to 9:30, 10:30 to 13:00, and 14:00 to 16:30 h. A matrix of functional traits was constructed for all species recorded in the field using the information obtained (Table 2).

The pollen load was also analyzed for the most abundant bee species. Species with an abundance greater than 5% were selected at each study site. Twenty individuals of each species were collected, and a sample of body pollen was taken from each using fuchsine-stained glycerinated gelatin (5%). For pollen identification, each sample was then placed on slides and analyzed. A palynology collection was established with pollen grains of arabica coffee. Samples were quantified until 300 pollen grains were obtained per slide, following the methodology proposed by [25,26]. The number of coffee pollen grains present in the sample was quantified for each site. The data obtained were used in the equation to estimate pollinator importance.

#### 2.2.3. Statistical Analyses

##### Taxonomic Diversity

For each study site and sampling time, the number of species and the number of individuals per species were determined. The species composition and abundance were described for each order. Chao 1 and abundance-based coverage estimator (ACE) estimates were obtained at a general level and for each sampling site for all flower-visiting insects as well as for the bee community using the statistical program EstimateS 9.1.0 [27]. Diversity profiles were estimated at a general level and per sampling site based on the matrix of abundances obtained in this study using the following indices: 

q = 0: richness; q = 1: exponential of the Shannon–Wiener diversity index; q = 2: the inverse of the Simpson concentration. 

The statistical packages iNEXT and SpadeR [28] were used to run the indices:Dq=(∑i=1Spiq)1/(1−q)
where

^q^D = diversity of order q;

S = number of species;

p_i_ = relative abundance (proportional abundance) of the ith species;

q = order of diversity (defines sensitivity to relative abundances of the species.

A descriptive analysis of the distribution of abundances at different sampling times was carried out.

##### Functional Diversity

Functional groups were defined for the bee community using a cluster analysis defined by Ward’s method. The functional space was graphed based on a distance matrix using non-metric multidimensional scaling.

The Gower distance matrix was constructed, and the following indices were estimated based on this matrix:

FDis = functional dispersion; FEve = functional evenness; FDiv = functional divergence. These indices were calculated using the FD-R package. The pollinator importance value (PIV) was estimated for each sampling site and bee species selected.
PIV = VR × PCC × C × PE
where VR = visit rate (relative abundance of the species with respect to the total sample); PCC = pollen-carrying capacity (proportion of *C. arabica* pollen found in the species with respect to all sampled bee species); C = flower constancy (mean proportion of *C. arabica* pollen per each species); PE = pollinator effectiveness (value assigned between 0 and 1, taking into consideration insect size and behavior within flowers, between flowers, and between plants).

## 3. Results

### 3.1. Taxonomic Diversity

A total of 23,735 individuals were recorded, distributed in 566 species, 105 genera, 84 families, and 10 orders (*Coleoptera*, *Dermaptera*, *Dictyoptera*, *Diptera*, *Hemiptera*, *Hymenoptera*, *Lepidoptera*, *Neuroptera*, *Orthoptera*, and *Thysanoptera*).

The *Hymenoptera* presented the highest abundance (22,528 individuals) and richness (215 species). Bees were the most abundant, with 20,838 individuals, belonging to 90 species of 4 of the 5 families present in Colombia *Apidae*, *Halictidae*, *Megachilidae*, and *Colletidae*, with the first two being the most representative in terms of number of species and individuals.

In the two coffee plots sampled at Pueblo Bello Experiment Station over a single flowering period for the main harvest, a total of 3834 individuals were recorded during the 90 h sampling effort; these insects belonged to 70 species, 15 families, and 5 orders (*Coleoptera*, *Dermaptera*, *Diptera*, *Hemiptera*, and *Hymenoptera*). The most abundant species were *A. mellifera* (2858 individuals), *Scaptotrigona* sp. 1 (180 individuals), *Tetragonisca angustula* (Latreille, 1811) (167 individuals), *Nannotrigona gaboi* Jaramillo, Ospina-Torres & Gonzalez, 2019 (160 individuals) (*Apidae*), *Ornidia obesa* (Fabricius, 1775) (Syrphidae) (8 individuals), and *Mordellidae* sp. 3 (*Coleoptera*) with 10 individuals.

Four plots were sampled at San Antonio Experiment Station during two flowering periods for the main harvest, and a total of 8359 individuals were recorded during the 180 h sampling effort. These insects belonged to 141 species, 37 families, and 7 orders (*Coleoptera*, *Dermaptera*, *Diptera*, *Hemiptera*, *Hymenoptera*, *Neuroptera*, and *Thysanoptera*). The more abundant species were *A. mellifera* (6126 individuals), *Geotrigona* cf. *tellurica* (1441 individuals) (*Hymenoptera*: *Apidae*), *Curculionidae* sp. 5 (*Coleoptera*) with 91 individuals, *Palpada* sp. 5 (*Diptera*: *Syrphidae*) with 22 individuals, and Miridae (*Hemiptera*) with six individuals.

Eight plots were sampled at Paraguaicito Experiment Station during two flowering periods for the main harvest and two for the mid-crop harvest, although several samplings were not carried out due to rains. A total of 4514 individuals were recorded during the 340 h sampling effort. These insects belonged to 233 species, 54 families, and 7 orders (*Coleoptera*, *Diptera*, *Dyctioptera*, *Hemiptera*, *Hymenoptera*, *Lepidoptera*, and *Thysanoptera*). The more abundant species were *A. mellifera* (188 individuals), *Lasioglossum* sp. 1, with 50 individuals (*Hymenoptera*: *Apidae*), *Dorymyrmex brunneus* Forel, 1908 (217 individuals), *Dorymyrmex biconis* Forel, 1912 (Formicidae) (148 individuals) (*Hymenoptera*: Formicidae), *Bibionidae* sp. 1 (27 individuals), *Diptera* sp. 2 (25 individuals), *Staphylinidae* sp. 5 (*Coleoptera*) with 15 individuals, and *Hesperiidae* (*Lepidoptera*) with 16 individuals.

Ten plots were sampled at the Naranjal Experiment Station during two flowering periods for the main harvest and two for the mid-crop harvest; a total of 4748 individuals were recorded during 450 h of sampling effort; these insects belonged to 288 species, 58 families, and 8 orders (*Coleoptera*, *Dermaptera*, *Diptera*, *Hemiptera*, *Hymenoptera*, *Lepidoptera*, *Neuroptera*, and *Orthoptera*). The more abundant species were corbiculated bees *A. mellifera* (1938 individuals), *T. angustula* (725 individuals), *Trigonisca* cf. *pediculana* (185 individuals), *Partamona* cf. *peckolti* (184 individuals), *Nannotrigona tristella* Cockerell, 1922 (138 individuals) (*Hymenoptera*: *Apidae*), *Bibionidae* sp. 2 (*Diptera*) with 17 individuals, and *Staphylinidae* sp. 2 (*Coleoptera*) presenting 33 individuals, considered outstanding.

Four plots were sampled at the Manuel Mejía Experiment Station during two flowering periods for the main harvest. A total of 2232 individuals were recorded during the 180 h sampling effort; these insects belonged to 99 species, 36 families, and 7 orders (*Coleoptera*, *Diptera*, Hemiptera, *Hymenoptera*, *Lepidoptera*, Orthoptera, and Thysanoptera). The more abundant species are corbiculated bees *A. mellifera* (1546 individuals), *Partamona* cf. *peckolti* (159 individuals), and *T. angustula* (157 individuals) (*Hymenoptera*: *Apidae*), *Astylus lebasi* Champion, 1918 (Melyridae) (21 individuals), *Staphylinidae* sp. 1 (26 individuals) (*Coleoptera*), and *Ornidia obesa* (*Diptera*: *Syrphidae*) with 10 individuals.

The number of species observed at all five sites was 566, which was lower than that estimated with two richness indicators: Chao1 and ACE (Table 3). According to the Chao1 richness estimator, the total number of species should be 874.9; in other words, the number of recorded species represents 64.7% of the expected value. According to ACE, the number should be 921.6, which means that the number of recorded species represents 61.4% of what was expected (Table 3). The number of species with a single individual, referred to as singletons, was 254.

Sampling coverage represents the fraction of total abundances represented in the sample. Although a relatively large fraction of species is missing based on study results, a general value of 0.989 is an indication that very few total individuals are still pending recording. Similarly, the sample coverage for each site was greater than 0.96 in all cases (Table 3), which suggests that it is unlikely that an individual collected in any future sampling will correspond to an unrecorded species.

For the bee community at a general level, the total number of species observed was 90, which is lower than that estimated with the Chao1 and ACE richness indicators (Table 4). The sampling of bees also presented a lower number of singletons (29), greater coverage, and a higher percentage of species represented in the sample (Table 4), as compared with the general analysis of flower visitors. Additionally, the confidence interval of the ACE estimator and that of the observed species overlap at all sites, but not in the general data, meaning that the observed value and the estimate value could be the same for each site.

The diversity of order q determines the influence that either common or rare species can have on the measure of diversity. The study results showed that, both at a general level and at the evaluation sites, the curve presents a steep slope with an increasing dominance of certain species (Figure 3). Hence, most of the species found at any of the evaluation sites were rare, and, at a general level, 357 species were singletons or doubletons, and 493 species did not exceed 10 recorded individuals. Four values conform to this measure of biodiversity: q = 0, when the value obtained is simply equivalent to species richness; q < 1, when rare species are overvalued; q = 1, when all species are included with a weight exactly proportional to their abundance in the community; q > 1, when more importance is given to abundant species [29].

The distribution of abundances in the different sampling times indicated that *A. mellifera* as well as insect species that do not correspond to the bee community presented a uniform distribution of abundance throughout the day, whereas native bees are more abundant in the time span from 10:30 to 13:00 h (Figure 4).

### 3.2. Functional Diversity

Five functional groups were obtained for the community of flower-visiting bees in coffee using a cluster analysis defined by Ward’s method (Figure 5). Based on the distance matrix, the group of corbiculate bees and non-corbiculate long-tongued bees occupied larger regions of the functional space and also presented greater abundances, whereas the functional groups formed by medium- and small-sized *Halictidae* and most of *Augochlora* sp.l individuals occupied small regions of the functional space (Figure 5).

The indices obtained using the Gower distance matrix showed that functional evenness (FEve), which describes the distribution of abundances within the functional space of the bee community, for the five sites presented values between 0.37 and 0.45 with an overall functional evenness of 0.41. This value is considered average because functional diversity can assume values between 0 and 1, where 1 is when all species are equally represented, and values are closer to 0 in the presence of species with very high-density values or of several high-abundance species whose functional values are very similar (Table 5).

Functional divergence (FDiv) presented values ranging between 0.93 and ~1.00 and a general value of 0.99. This index ranges between 0 and 1 and measures the distribution of the abundances of species within the functional space, i.e., the most abundant species are distributed in different regions of the functional space (Table 5, Figure 5).

The functional dispersion (FDis) for the entire bee community was 0.19, and for sites, it ranged between 0.10 and 0.27 (Table 5). This index estimates the average distance of the species from the centroid of the community, which is influenced by the abundances of each species. This index presents a lower limit of 0 and has no upper limit. Therefore, its values can be compared only with those of other studies; however, the values obtained across sites were similar.

Bee species presenting abundances greater than 5% were selected to analyze pollen load, several of which are common across study sites (Table 6, Figure 6).

The pollination importance value (PIV) estimated for each species was found to depend on the collection site. However, for most of the species evaluated, the C value was greater than 0.9 and the PCC was greater than 0.3. The species presenting the lowest C values were *Lasioglossum* spp. and *Partamona* cf. *peckolti* (Table 6).

Species such as *A. mellifera* presented a PE score of 1, which was the highest score, attributable to its behavior when visiting the coffee flower, making frequent contact with stigma and stamens when looking for resources other than pollen. Bee species such as *Partamona* cf. *peckolti*, *Scaptotrigona* sp. 1, *T. angustula* at most sites, and *Geotrigona* cf. *tellurica* presented values higher than 0.92 (Table 6), considered high PE values. Meanwhile, *T. angustula* at the Naranjal Experiment Station presented a value of 0.67 and *Lasioglossum* spp. one of 0.65, given that no frequent stigma contact was observed and bees did not touch the stamens when looking for resources other than pollen.

*Apis mellifera* presented the highest relative abundance of species (VR) value, ranging between 70.62 and 73.62. The VR is important because it is one of the measurements that reduces the PIV value for infrequent flower visitors because, as mentioned above, the dominance of a few species makes this value very low for the others. *Apis mellifera* presented the highest PIV, scoring 59.13 at the Paraguaicito Experiment Station, while *Lasioglossum* spp. presented the lowest, 0.11, at the same site (Table 6).

## 4. Discussion

This study reported 566 species of flower-visiting insects in coffee crops, 90 of which correspond to bee species (Appendix A, Table A1). To date, this is the highest number of species of bees and other flower-visiting insects in coffee reported by any study in the literature (Appendix B, Table A2).

For the first time, bees of the genera *Agapostemon*, *Augochloropsis*, *Augochlorella*, *Caenohalictus*, *Habralictus*, *Pereirapis*, *Pseudaugochlora*, and *Ptiloglossa* are reported as coffee flower visitors, as well as the species *Bombus pauloensis* Friese, 1912, *Bombus melaleucus* Handlirsch, 1888, *Eulaema cingulate* (Fabricius, 1804), *Geotrigona kaba* Gonzalez & Sepúlveda, 2007, *Geotricona* cf. *tellurica*, *Nannotrigona gaboi*, *N. pilosa* Jaramillo, Ospina & Gonzalez, 2019, *N. tristella* (Cockerell 1922), *Nasutopedia* sp., *Paratrigona opaca* (Cockerell, 1917), *Tetragona perangulata* (Cockerell, 1917), *Trigona fulviventris* Guérin-Méneville, 1844, *Trigonisca* cf. *mepecheu*, and *Trigonisca* cf. *pediculana* (*Hymenoptera*: *Apidae*), the genera *Agelaia*, *Angiopolybia*, *Epipona*, *Pachynoderus*, *Parachartergus*, *Polistes*, *Polybia*, *Protopolybia*, and *Synoeca* (*Hymenoptera*: *Vespidae*), *Ornidia major* (Curran 1930), and *Pseudodoros clavatus* (Fabricius, 1794) (*Diptera*) were also identified here, which complement the results of [9].

The present study reports a greater richness of coffee flower visitors as compared with previous studies. However, based on the Chao1 and ACE estimators, there are still between 35.3 and 38.6% of species to be collected. Study sites presenting the lowest species richness were Pueblo Bello, San Antonio, and Manuel Mejía experiment stations. The sampling coverage for all sites was greater than 97%, which means that there are only a few individuals pending collection. Therefore, the sampling coverage of this study is considered complete; according to [30], the coverage is calculated considering the abundances and is defined as the probability that the next individual collected belongs to a species already represented in the sample. Therefore, the probability of finding coffee flower-visiting individuals of species not represented in the sample is low.

The reason why sampling coverage is considered complete, that is, with values close to 1, is because only a few species were hyper-abundant, making other species considered rare. Consequently, 254 singletons were found in the general sample, while for each of the study sites, it ranged between 36 and 145. *Coleoptera*, *Diptera*, and Hemiptera were those with the greatest number of singletons; in many cases, the species of these orders do not include flowers in their diets to feed their young [31]. It could also be attributed to the fact that the abundance and richness of *Coleoptera* and *Lepidoptera* were not fully sampled, according to Knop et al. [32], these flower visitors are more active at night. The representation of these groups was also low in studies carried out by [13,33,34,35].

The bee community was better represented than the general insect community. The percentage of species represented in the sample ranged between 47.7 and 88.9%, depending on the estimator used and the study site. The sampling coverage was between 0.995 and 0.999, and the number of species represented by a single individual was 29, which is equivalent to 32.2% (Table 2). In most studies conducted on flower visitors to coffee and other crops, bees were the most abundant, attributable to their life history traits and the role they play as pollinators [22,36].

In most of the studies that have evaluated flower visitors in the coffee species *C. arabica* and *C. canephora*, bees were the most abundant insect species, particularly social bees of the tribes Apini and Meliponini [9]. This agrees with the results of this study, where *A. mellifera* was the most abundant species, followed by species of the genera *Geotrigona*, *Nannotrigona*, *Partamona*, *Scaptotrigona*, and *Tetragonisca*. In the case of social bees, abundance may be related to the availability of a specific flower resource because these bees preferably forage in the most abundant flower resources due to the use of a recruitment system [37]. In the study carried out by Bänsch et al. [38] on strawberries, results indicated that solitary bees may be excluded from concentrated blooms because social bees compete for this type of resource. Solitary bees are easier to use in crops with scattered blooms, which explains why social bees were the most abundant species in all sites sampled in this study because coffee is characterized by having few abundant blooms per year, depending on the region.

The diversity profiles of the order q (Figure 2) indicate that only a few species have high dominance in the community of flower-visiting insects in coffee, which implies a low functional evenness. This behavior could have implications for the community because, according to Hillebrand et al. [39], the high dominance of species can reduce ecosystem function as it not only reduces the stability of the pollination process but also makes the ecosystem more vulnerable to invasion by alien species.

Klein et al. [40] found that pollination in coffee depends mainly on the diversity of flower visitors, and, according to Moritz et al. [41], although *A. mellifera* seems to have no effect on species’ richness at the local or regional levels, it could be responsible for species dominance in the evaluated communities. Social bees could therefore be of interest for the pollination of coffee. To enhance the contribution to coffee production, the strategy forward should focus on increasing the abundance not only of social species other than *A. mellifera* but also that of solitary bees to better conserve the ecosystems surrounding coffee plantations.

The results from this study showed that native bee species are most abundant between 10:30 and 13:00, whereas the abundance of *A. mellifera* and insect species of the other taxonomic groups was similar during the three sampling periods (7:00 to 9:30, 10:30 to 13:00, and 14:00 to 16:30). According to [32,42,43], the activity of daytime flower visitors is determined by variables such as sunlight and temperature, not by a specific hour.

The functional evenness (FEve) value for the bee community, both at a general level and for the study sites, was around 0.4, which suggests that species of the flower-visiting bee community in coffee are unevenly distributed in the functional space (Figure 5). Because several species present overlapping traits and there are empty spaces between them, these empty spaces are referred to as unoccupied niches, which could be inhabited by other flower-visiting insects or by invasive species in coffee crops. Empty spaces also imply that the ecosystem function is not productive or reliable because, in the event of a disturbance, hyperabundant species (those currently responsible for most of the pollination in coffee) could be affected [44,45].

On the other hand, functional divergence presented values > 0.9, indicating that the most abundant species, regardless of the locality, presented non-overlapping features, meaning that these species occupy different regions of the functional space [46]. Therefore, the most abundant bee species found visiting coffee flowers are dissimilar and compete weakly with each other. Taking Paraguaicito Experiment Station as an example, at this study site, the most abundant species were *A. mellifera*, *Trigonisca* cf. *pediculana*, and several species of the genus *Lasioglossum*, which showed differences in nesting habits, size, pollen-carrying structure, and times of greatest abundance. This is important for pollination in coffee crops because it explains why coffee flower visitors, some of which are pollinators, present non-overlapping traits and why the pollination function could be resistant to disturbances [47].

Functional dispersion is relevant in different crops. Martins et al. [48], for example, found that, in apples, greater functional dispersion translated into a higher fruit set. This could also be the case with coffee. The present study found functional dispersion values that ranged between 0.10 and 0.27, with a mean of 0.17, which is considered medium-low. In studies carried out by [48,49] in areas with different degrees of grazing intensity, functional dispersion values ranged between 0.10 and 0.45 for the first study and between 0.23 and 0.32 for the second, which means that fruit set in coffee could increase with increasing functional dispersion. Consequently, Gómez et al. [50] found that the *C. arabica* species, despite being a self-pollinated plant, benefits from the presence of flower-visiting insects in the crop, which contributed 16.3% to berry set, 26.9% to yield, and 30.6% to the physical quality of coffee.

This study did not address the functional diversity of the entire community of flower-visiting insects in coffee, only that of the bee community. As a result, empty spaces appeared when the functional space was graphed, the hypothesis being that these could be occupied by insects similar to bees but with different traits, and were therefore not taken into account in the analysis.

The species *A. mellifera* presented the highest abundance as well as the highest PCC, with values between 0.33 and 0.90 depending on the study site. It also presented a high C value (0.94–0.98) and, as a result, the highest PIV as well. According to Roubik [15], *A. mellifera* is probably one of the best pollinators in coffee crops, which could be because this species is native to Africa, the center of origin of *C. arabica*. However, native bees such as *N. gaboi*, *T. angustula*, *Geotrigona* cf. *tellurica*, and *Partamona* cf. *peckolti*, despite being small bees, presented not only higher PCC values but also greater C constancy. However, their PIV was lower not only because they presented lower abundances than *A. mellifera* but also because their PE value was lower as they presented little stigma contact.

This leads us to understand that, although an individual carries large and exclusive amounts of pollen, this does not necessarily make it a good pollinator, because if it comes into contact with male structures and not female structures, it will not fulfill the pollination function. Although it is important to define the entire community of flower visitors to avoid biases [51], it is equally important to directly measure pollen deposition in stigmas and fruit sets for each species, as pollen load or percentage of visits are not necessarily good indicators of pollination efficiency [52]. Despite the above, these secondary species have the potential to be good pollinators for coffee crops and should be addressed in future studies.

## 5. Conclusions

In Colombia, coffee flowers are visited by 566 insect species from different taxonomic groups, but this number could be even higher according to the Chao1 and ACE estimators. The group of bees was the most abundant and diverse, with 89 native species being recorded, all natural inhabitants of Colombian coffee ecosystems and distributed throughout the country. These bees represent 16% of the bee species described for Colombia. This study reports eight genera of bees that had not been previously reported in coffee crops, as well as 14 species of the family *Apidae*. It is also the first study to address the functional diversity of bees visiting coffee crops, and the PIV is determined to select potential pollinator species for *C. arabica.*

Based on functional diversity indices, the community of flower-visiting bees in coffee crops may have low resistance to disturbance and temporary instability. The performance of the pollination service could also be better.

When implementing a strategy to increase coffee production using insect-based pollination services, the approach should aim at (1) increasing the abundances of the most efficient and effective species in pollination, such as the native social bees *N. gaboi*, *T. angustula*, *Geotrigona* cf. *tellurica*, and *Partamona* cf. *peckolti*. (2) strengthen the complementary between species favoring the abundance of the functional groups of the small and medium-sized *Halictidae*, and (3) strengthen facilitation through the presence of *Diptera* (Syrphidae) that contribute to maintaining floristic diversity, as several species could facilitate the permanence or role of others as pollinators of a given crop, and a better understanding of the taxonomic diversity of flower visitors can serve to successfully adjust the estimates of insects present in a given region or crop.

Crop management strategies should be proposed that integrate pollinators as a major component and focus on balancing abundances for several bee species, considering the hours of greatest activity (between 10:30 and 13:00).

Future studies should also address the direct measurement of pollination, such as pollen deposition on stigmas and fruit set, in species identified with the greatest potential to serve as pollinators in coffee crops.

## Figures and Tables

**Figure 1 insects-15-00143-f001:**
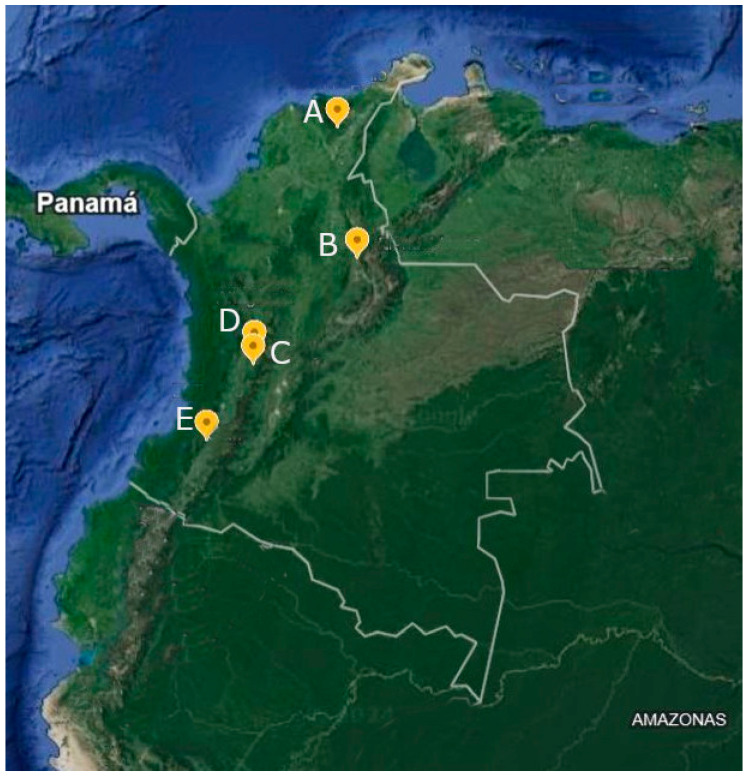
Study sites located in Colombia coffee-growing region. (A) Pueblo Bello Experiment Station; (B) San Antonio Experiment Station; (C) Paraguicito Experiment Station; (D) Naranjal Experiment Station; and (E) Manuel Mejía Experiment Station.

**Figure 2 insects-15-00143-f002:**
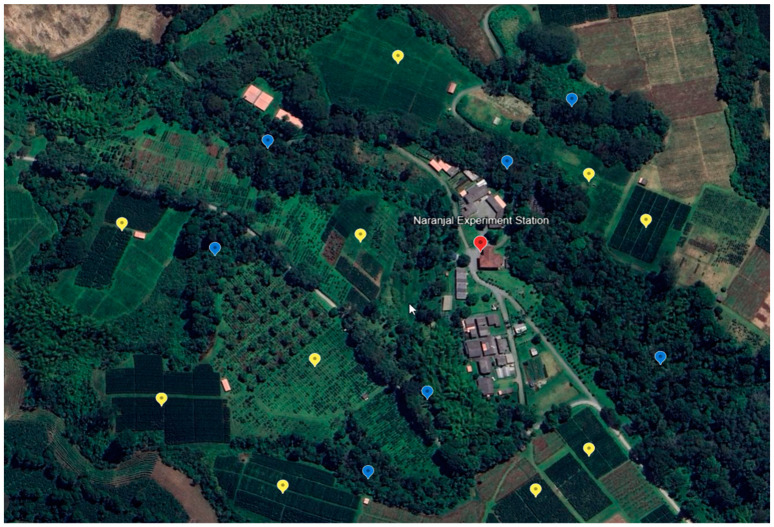
Naranjal Experiment Station. Yellow marker (coffee plots). Blue marker (natural vegetation and secondary forest).

**Figure 3 insects-15-00143-f003:**
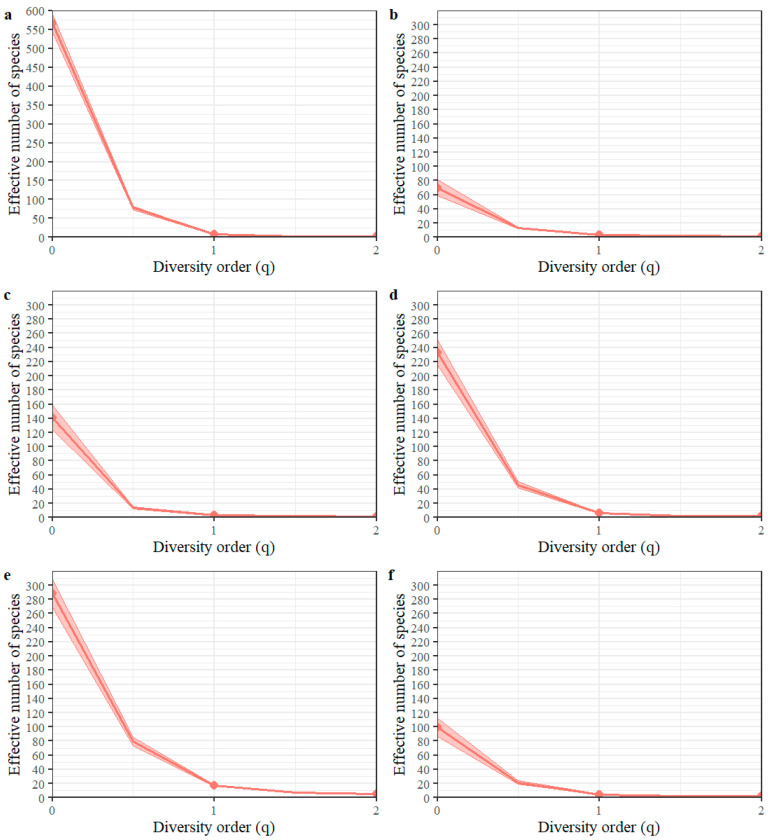
Diversity profiles of order q for (**a**) general; (**b**) Pueblo Bello Experiment Station; (**c**) San Antonio Experiment Station; (**d**) Paraguaicito Experiment Station; (**e**) Naranjal Experiment Station; and (**f**) Manuel Mejía Experiment Station.

**Figure 4 insects-15-00143-f004:**
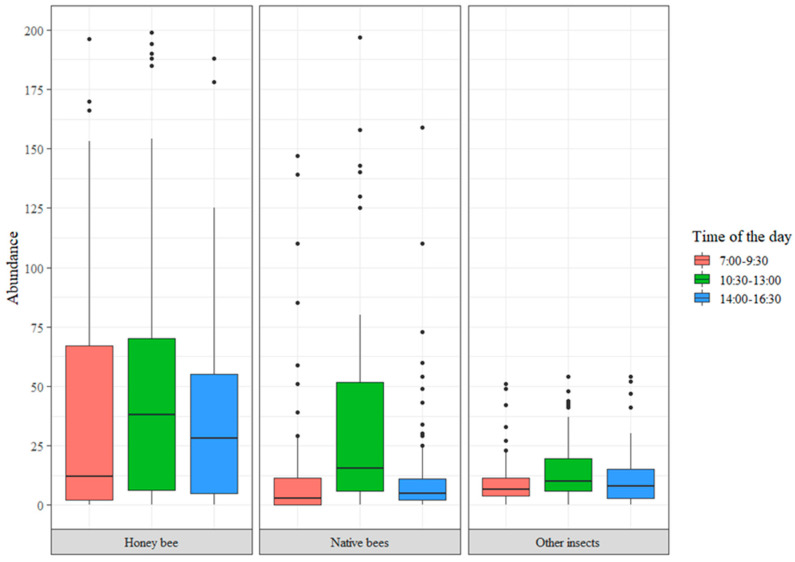
Distribution of abundances at different sampling times, where the central line is the median and boxes and vertical lines represent the quartiles.

**Figure 5 insects-15-00143-f005:**
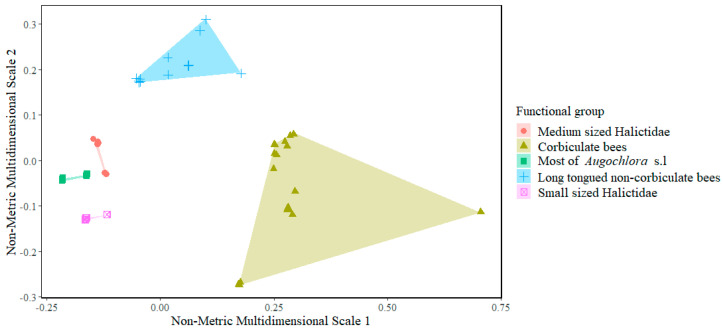
Functional groups for the flower-visiting bee community in coffee and representation of the functional space for each group, obtained by non-metric multidimensional scaling.

**Figure 6 insects-15-00143-f006:**
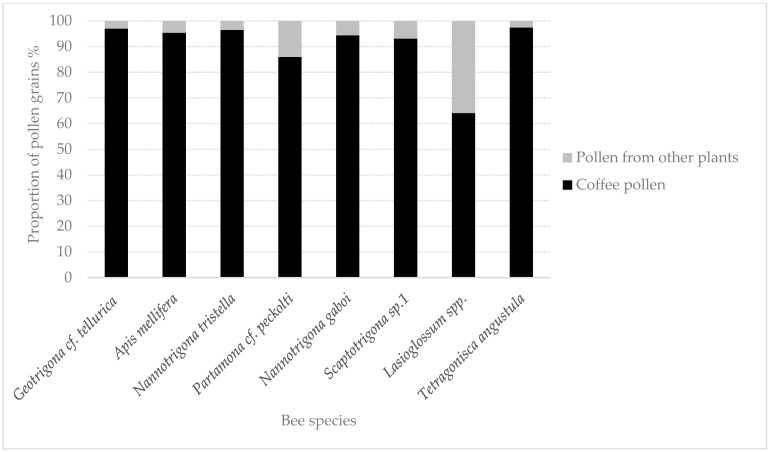
Pollen load for the most abundant and frequent bee species flower-visiting in coffee crops.

**Table 1 insects-15-00143-t001:** Description of study sites located in Colombia coffee-growing region [18].

Study Sites	Location	Climate Conditions	Harvest Distribution	Area Cultivated in Coffee (Hectares)
(A) Pueblo Bello Experiment Station	Municipality of Pueblo Bello (department of Cesar), located in the Sierra Nevada de Santa Marta mountain range (10°25′ N; 73°34′ W); altitude: 1134 m above sea level; coffee ecotope: 402.	Average temperature of 27.9 °C; mean relative humidity of 78.3%; annual precipitation of 1727.7 mm; and 2302 h of sunshine/year.	Main harvest (90%): flowering between March and April; mid harvest (10%): flowering between August and September.	25.6
(B) San Antonio Experiment Station	Municipality of Floridablanca (department of Santander), Cordillera Oriental mountain range, western slope (07°06′ N; 73°04′ W); altitude: 1539 m above sea level; coffee ecotope: 302A.	Average temperature of 20.1 °C; average RH of 79.4%; annual precipitation of 1644 mm; 1155 h of sunshine/year.	Main harvest (90%): flowering between March and April; mid harvest (10%): flowering between August and September.	3.22
(C) Paraguaicito Experiment Station	Municipality of Buenavista (department of Quindío), Cordillera Central mountain range, western slope (04°4′ N; 75°44′ W); 1303 m above sea level; coffee ecotope: 211A.	Average temperature of 22.4 °C; average RH of 78.4%; annual precipitation of 1938 mm; 1541 h of sunshine/year.	Main harvest (55%): flowering between February and March; mid harvest (45%): flowering between August and September	16.3
(D) Naranjal Experiment Station	Municipality of Chinchiná (department of Caldas), Cordillera Central mountain range, western slope (4°58′ N; 75°39′ W); 1381 m above sea level; coffee ecotope: 206A.	Average temperature of 21.6 °C; average RH of 80.6%; annual precipitation of 2990 mm; 1537 h of sunshine/year.	Main harvest (75%): flowering between January and March; mid harvest (25%): flowering between August and September.	48.7
(E) Manuel Mejía Experiment Station	Municipality of El Tambo (department of Cauca), Cordillera Central, western slope (02°24′ N; 76°44′ W); altitude: 1735 m above sea level; coffee ecotope: 218A.	Average temperature of 19.8 °C; average RH of 81.1%; annual precipitation of 1826 mm; 1632 h sunshine/year.	Mid harvest (10%): flowering between February and March; main harvest (90%): flowering between August and September	10.9

**Table 2 insects-15-00143-t002:** Functional traits selected to characterize the functional diversity of the bee community.

Trait	Status of Trait	Type of Variable
Nesting habit	Pre-existing cavities in the ground, pre-existing cavities, new cavities in the ground, tree branches, decaying wood, dry logs and wood, and dry branches.	Nominal categorical
Degree of sociability	Eusocial, parasocial, solitary (subsocial and solitary), facultative parasocial, and facultative eusocial.	Nominal categorical
Size (mm)	Mean of the intertegular distance for collected individuals; maximum 20 individuals measured.	Continuous numeric
Stigma contact	Very frequent; always; occasional; never.	Nominal categorical (numeric for the FD package)
Location of pollen loading structure	Tibial, femorotibial, basitarsotibial, and gasteral.	Nominal categorical
Type of tongue	Short, long, and very long.	Ordinal categorical
Abundance from 7:00 to 9:30	Number of individuals.	Whole numbers
Abundance from 10:30 to 13:00	Number of individuals.	Whole numbers
Abundance from 14:00 to 16:30	Number of individuals.	Whole numbers

**Table 3 insects-15-00143-t003:** Estimate of number of species and sampling coverage of flower-visiting insects in coffee crops per study site and in general.

Study Sites	Species Observed	Singletons	Sampling Coverage	Expected Species		Percentage Species in Sample	
				Chao1	ACE	Chao1	ACE
Pueblo Bello E.S *	70	36	0.991	133	155.8	52.6	44.9
San Antonio E.S	141	99	0.988	679.9	510.1	20.7	27.6
Paraguaicito E.S	233	125	0.973	405.2	439.2	57.5	53
Naranjal E.S	288	145	0.970	496.8	529.3	58.0	54.4
Manuel Mejía E.S	99	61	0.973	302.2	277.0	32.8	35.7
General	566	254	0.989	874.9	921.6	64.7	61.4

* E.S = experiment station.

**Table 4 insects-15-00143-t004:** Estimate of the number of species and sampling overage of the bee community per site and in general.

Study Sites	Observed Species	Singletons	Sampling Coverage	Expected Species		Percentage Species in Sample	
				Chao1	ACE	Chao1	ACE
Pueblo Bello E.S *	30	11	0.997	48.3	55.5	62.1	54.1
San Antonio E.S	23	9	0.999	35.0	48.2	65.7	47.7
Paraguaicito E.S	33	10	0.997	37.1	41.1	88.9	80.3
Naranjal E.S	51	18	0.995	66.3	71.1	76.9	71.7
Manuel Mejía E.S	20	7	0.996	27.0	28.2	74.1	70.9
General	90	29	0.999	119	125.3	75.6	71.8

* E.S = experiment station.

**Table 5 insects-15-00143-t005:** Functional diversity indices calculated for the flower-visiting bee community in coffee crops per site and in general.

Study Sites	Functional Species	FEve ^a^	FDiv ^a^	FDis ^a^
Pueblo Bello E.S *	30	0.37	0.98	0.18
San Antonio E.S	23	0.39	0.93	0.16
Paraguaicito E.S	33	0.43	~1.00	0.10
Naranjal E.S	51	0.39	0.99	0.27
Manuel Mejía E.S	20	0.45	~1.00	0.16
General	90	0.41	0.99	0.19

* E.S = experiment station. ^a^ FEve = functional evenness; FDiv = functional divergence; FDis = functional dispersion.

**Table 6 insects-15-00143-t006:** Estimated values of different indices for each site and bee species.

Study Sites	Species	VR(A) ^a^	PCC ^a^	C ^a^	PE ^a^	PIV ^a^
Pueblo Bello E.S	*A. mellifera*	73.62	0.47	0.98	1.00	33.51
	*N. gaboi*	4.12	0.53	0.94	0.92	1.91
	*T. angustula*	4.30	0.47	0.99	0.67	1.33
	*Scaptotrigona* sp. 1	4.64	0.44	0.93	0.92	1.77
San Antonio E.S	*A. mellifera*	73.29	0.47	0.95	1.00	32.81
	*Geotrigona* cf. *tellurica*	17.24	0.53	0.97	0.92	8.10
Paraguaicito E.S	*A. mellifera*	70.62	0.90	0.93	1.00	59.13
	*Lasioglossum* spp.	2.64	0.10	0.64	0.65	0.11
Naranjal E.S	*A. mellifera*	40.82	0.39	0.98	1.00	15.62
	*N. tristella*	2.91	0.22	0.97	0.87	0.53
	*T. angustula*	15.27	0.39	0.99	0.67	3.93
Manuel Mejía E.S	*A. mellifera*	69.27	0.33	0.94	1.00	21.75
	*Partamona* cf. *peckolti*	7.12	0.35	0.86	0.93	2.00
	*T. angustula*	7.03	0.31	0.95	0.67	1.41

E.S = Experiment station. VR (A) ^a^ = estimated visit rate (relative abundance); PCC ^a^ = pollen-carrying capacity; C ^a^ = flower constancy; PE ^a^ = pollinator effectiveness; and PIV ^a^ = pollination importance value.

## Data Availability

All data are contained within the article or Appendix A.

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
