# Peer review of "Taxonomic and Functional Diversity of Flower-Visiting Insects in Coffee Crops"

_insects, 2024, doi:10.3390/insects15030143_

Round 1

Reviewer 1 Report

Comments and Suggestions for Authors

Line 21: "no studies have been carried out" is contradictory to what you say in line 73. I think you mean to say that of the previous studies of coffee pollinators in Columbia, none of them have incorporated functional diversity into their analysis.

Line 23 (and others): when you describe the experimental design, you should say something along the lines of "were selected among five sites..." This would clear up the interpretation that there are 28 plots total - not 28 for every site. Also, there are some instances (especially later in the manuscript) where the language for site and plot are not consistent.

Line 43: "insect richness and density are the most important crop yield predictors" this requires a citation to support this claim.

Line 55: suggest edit "Additionally, functional diversity provides valuable..."

Line 68: There is no detail explaining what you mean by "most important..." Later in the manuscript you make the distinction between abundance, richness, and diversity to possible increases to crop yield. This is a major gap in your argument in this section.

Line 81: suggest edit - change "presenting" to "representing"

Line 87: this is further evidence for the unclear claim in the abstract that there are no other studies of coffee pollination in Colombia.

Line 92: suggest edit "corroborated, indicating a community..."

Line 96: should be focuses for correct present tense when discussing the current manuscript

Table 1: I think the reference column is superfluous since the article you reference is the same for each study site. You could mention the article in the table caption. Additionally, I would suggest that you label each study site as "A", "B", "C", etc. and keep the same order as what is mentioned in the caption of figure 1. Lastly, there is no mentioning of the relative size of some of these studies. I would suggest mentioning the relative distances between the different plots located within each study site.

Line 114: "as follows:"

Line 117: "were grown"

Line 118: it is not clear what you mean by "contained shade" were the plants growing under shade? Do you think this would impact insect pollinator visitation? Connect those dots for the reader.

Line 130: another example of inconsistent language used to describe your experimental design

Line 153: seems like a subheading. I would suggest rewording.

Line 155: this is the first mentioning of the ACE acronym, and you need to move the explanation of the acronym to here (not lien 244)

Line 169: uses different font size

Line 180-186: the description of this PIV equation needs more details, and I would suggest walking through a species/specimen as an example. Specific details are needed for the more subjective variables like Pollinator Effectiveness

Line 197: crops is another example of inconsistent terminology

Line 243: the term general doesn't seem to fit. Do you mean total numbers among all five sites? Total is more appropriate than general since you are summing the values over the five sites.

Table 3: If the table is going to span multiple pages, then the headings should be repeated. This might be a comment to the person formatting the final version of the manuscript.

Line 261: this description is repeated from the previous section. Unless you strongly argue that this provides value to the reader, I would remove it.

Line 306: how do these values compare to other systems? Compare to other studies in Colombia or South America? Compare to other studies in coffee pollination.

Table 5: richness and functional species are the same, so why include both?

Table 6: also need to repeat headers on next page.

Line 370-373: this is a run-on sentence. Needs rewording

Lien 380-383: it is not clear what you are trying to say here. Needs rewording

Line 421: suggest edit, "These results showed that..." or "The results from this study showed..."

Line 430: the unoccupied niche space could be from the species not collected using your methods such as species active from 7:00-16:30

Line 448: is the only mentioning of crop yield. This is a crucial part of your argument

Line 466: you mention some difference in PE values among the species, but there is no context to how much a change in said value is relevant in this system. Is a 0.01 change in the value relevant? 0.1?

Comments on the Quality of English Language

Some minor issues with grammar. See other comments.

Author Response

Thank you for the valuable corrections and suggestions to improve the paper

Specific comments:

Line 21: "no studies have been carried out" is contradictory to what you say in line 73. I think you mean to say that of the previous studies of coffee pollinators in Columbia, none of them have incorporated functional diversity into their analysis.

Answer: Accepted, the sentence was rewritten as follows: “Despite the important role that flower-visiting insects play in agricultural production, of the previous studies of coffee pollinators in Colombia, none of them have incorporated functional diversity into their analysis. Therefore, this study aimed to quantify the abundance, richness and functional diversity of insects that visit flowers in coffee crops”.  

Line 23: (and others): when you describe the experimental design, you should say something along the lines of "were selected among five sites..." This would clear up the interpretation that there are 28 plots total - not 28 for every site. Also, there are some instances (especially later in the manuscript) where the language for site and plot are not consistent.

Answer: Accepted, the sentence was rewritten as follows: “Twenty-eight plots were selected among five sites located in the north, center and south of Colombia

Line 43: "insect richness and density are the most important crop yield predictors" this requires a citation to support this claim.

Answer: Accepted, the reference was added

Line 55: suggest edit "Additionally, functional diversity provides valuable..."

Answer: Accepted, the sentence was rewritten as follows: “Additionally, functional diversity provides valuable understanding of the role of species and allows the incorporation of characteristics that influence their performance and processes in the ecosystem [5]. In addition, it allows us to know the response of species to climate change” [6].

Line 68: There is no detail explaining what you mean by "most important..." Later in the manuscript you make the distinction between abundance, richness, and diversity to possible increases to crop yield. This is a major gap in your argument in this section.

Answer: Accepted, the sentence was rewritten as follows: ´´were the most important  in the  pollination of C. arabica´´

Line 81: suggest edit - change "presenting" to "representing"

Answer: Accepted

Line 87: this is further evidence for the unclear claim in the abstract that there are no other studies of coffee pollination in Colombia.

Answer: Accepted, the change was made in the abstract

Line 92: suggest edit "corroborated, indicating a community..."

Answer: Accepted

Line 96: should be focuses for correct present tense when discussing the current manuscript

Answer: Accepted

Table 1: I think the reference column is superfluous since the article you reference is the same for each study site. You could mention the article in the table caption. Additionally, I would suggest that you label each study site as "A", "B", "C", etc. and keep the same order as what is mentioned in the caption of figure 1. Lastly, there is no mentioning of the relative size of some of these studies. I would suggest mentioning the relative distances between the different plots located within each study site.

Answer: Accepted, The table and figure were adjusted

Line 114: "as follows:"

Answer: Accepted

Line 117: "were grown"

Answer: Accepted

Line 118: it is not clear what you mean by "contained shade" were the plants growing under shade? Do you think this would impact insect pollinator visitation? Connect those dots for the reader.

Answer: Accepted, the sentence was rewritten as follows: ´´All coffee crops were grown in places exposed to the sun, except at the Pueblo Bello Experimental Station, where coffee was grown in the shade´´.

Line 130: another example of inconsistent language used to describe your experimental design

Answer: Accepted, the sentence was rewritten as follows: ´´With the registered information, for each study site and samplings time, the number of species and the number of individuals per species was determined ´´.

Line 153: seems like a subheading. I would suggest rewording.

Answer: Accepted, the sentence was rewritten as follows: “For each study site and samplings time, the number of species and the number of individuals per species was determined´´.

Line 155: this is the first mentioning of the ACE acronym, and you need to move the explanation of the acronym to here (not lien 244)

Answer: Accepted

Line 169: uses different font size

Answer: Accepted

Line 180-186: the description of this PIV equation needs more details, and I would suggest walking through a species/specimen as an example. Specific details are needed for the more subjective variables like Pollinator Effectiveness

Line 197: crops is another example of inconsistent terminology

Answer: Accepted

Line 243: the term general doesn't seem to fit. Do you mean total numbers among all five sites? Total is more appropriate than general since you are summing the values over the five sites.

Answer: Accepted, the sentence was rewritten as follows: “The number of species observed all five sites was 566”

Table 3: If the table is going to span multiple pages, then the headings should be repeated. This might be a comment to the person formatting the final version of the manuscript.

Answer: Accepted, the table was adjusted

Line 261: this description is repeated from the previous section. Unless you strongly argue that this provides value to the reader, I would remove it.

Answer: Accepted, the sentence was removed

Line 306: how do these values compare to other systems? Compare to other studies in Colombia or South America? Compare to other studies in coffee pollination.

Table 5: richness and functional species are the same, so why include both?

Answer: Accepted, the richness was removed

Table 6: also need to repeat headers on next page.

Answer: Accepted

Line 370-373: this is a run-on sentence. Needs rewording

Answer: Accepted, the sentence was rewritten as follows:  “Therefore, the sampling coverage of this study is considered complete; according to [30] the coverage is calculated considering the abundances and is defined as the probability that the next individual collected belongs to a species already represented in the sample.

Lien 380-383: it is not clear what you are trying to say here. Needs rewording

Answer: Accepted, the sentence was rewritten as follows:  “The reason why sampling coverage is considered complete, that is, with values close to 1, is because only a few species were hyper-abundant, making other species be considered as rare. Consequently, 254 singletons were found in the general sample, while for each of the study sites it ranged between 36 and 145. Coleoptera, Diptera and Hemiptera were those with the greatest number of singletons, in many cases the species of these orders do not include flowers in their diets to feed their young [31].”

Line 421: suggest edit, "These results showed that..." or "The results from this study showed..."

Answer: Accepted

Line 430: the unoccupied niche space could be from the species not collected using your methods such as species active from 7:00-16:30

Answer: It may be, but there is no support to affirm it.

Line 448: is the only mentioning of crop yield. This is a crucial part of your argument

Answer: Accepted, the following sentence was added “Consequently Gómez et al. [50] found that the C. arabica species, despite being a self-pollinated plant, benefits from the presence of flower-visiting insects in the crop, which contributed 16.3% to berry set, 26.9% to yield, and 30.6% to the physical quality of coffee”.

Line 466: you mention some difference in PE values among the species, but there is no context to how much a change in said value is relevant in this system. Is a 0.01 change in the value relevant? 0.1?

Reviewer 2 Report

Comments and Suggestions for Authors

The authors present an interesting inventory of insects visiting coffee flowers in Columbia. Such inventory still needs to be published. Observations were conducted on several sites across the country at different times of the year and the day, resulting in an impressive 1240 hours of observation and 23 735 individuals recorded, belonging to 566 species.

The authors analyse functional diversity based on several parameters, including behaviour (stigma contact), abundance in different seasons, and pollen load. Based on their observations and previous literature, the authors analyse the insect diversity regarding their potential interest in providing pollination services to coffee crops.

This study is well explained and presented, and it provides an interesting and new overview of flower-visiting insects in coffee crops that could help understand and manage coffee pollination. Nevertheless, it could help the readers understand the coffee crops' environment well by better describing the study fields. Moreover, despite what the authors say, they still need to provide all the data they recorded and discussed. They need to provide a full table of the species recorded (maybe including site, date and time of observation) and a table of the pollen load observations.

Please find some more specific remarks below:

Fig 1 A scale needs to be included; please add it. I assume the experimental stations do not consist of the entire areas in dark green. Please add a cross or a dot to localise the stations more precisely.

L 112-124. What is the size of plots (mean size can be sufficient). How far are they apart? I am not that familiar with coffee crops. How dense is the plantation? Is the management the same between each station? (Organic? Density? Age? Environment? (Close to forest, including other trees in the fields, grass or other crops on the soil?)( It would be nice to present more of the coping system to provide clues to what could be more or less of interest for different insect species.

L 169 issues with font size.

L 182, 183, 219, 220, 330, 334, 343 (and others) check that Latin names are in italics.

It would be nice to provide a second appendix with the complete list of recorded insects in the present study. (Maybe including data such as full name, family, number of individuals, number of sites, season, and daytime?)

Providing data about pollen observation is needed, especially as you provide no figures about it.

L 516 "all data are contained within the article or the Appendix A1." That is not yet true. So please provide all the data, as a complete list of species recorded, pollen observation, etc…

References check that Latin names are in italics.

Author Response

Thank you for the valuable corrections and suggestions to improve the paper

Specific comments:

The authors present an interesting inventory of insects visiting coffee flowers in Columbia. Such inventory still needs to be published. Observations were conducted on several sites across the country at different times of the year and the day, resulting in an impressive 1240 hours of observation and 23 735 individuals recorded, belonging to 566 species.

The authors analyse functional diversity based on several parameters, including behaviour (stigma contact), abundance in different seasons, and pollen load. Based on their observations and previous literature, the authors analyse the insect diversity regarding their potential interest in providing pollination services to coffee crops.

This study is well explained and presented, and it provides an interesting and new overview of flower-visiting insects in coffee crops that could help understand and manage coffee pollination. Nevertheless, it could help the readers understand the coffee crops' environment well by better describing the study fields. Moreover, despite what the authors say, they still need to provide all the data they recorded and discussed. They need to provide a full table of the species recorded (maybe including site, date and time of observation) and a table of the pollen load observations.

Please find some more specific remarks below:

Fig 1 A scale needs to be included; please add it. I assume the experimental stations do not consist of the entire areas in dark green. Please add a cross or a dot to localise the stations more precisely.

Answer: Accepted, Figure 1 was changed

L 112-124. What is the size of plots (mean size can be sufficient). How far are they apart? I am not that familiar with coffee crops. How dense is the plantation? Is the management the same between each station? (Organic? Density? Age? Environment? (Close to forest, including other trees in the fields, grass or other crops on the soil?)( It would be nice to present more of the coping system to provide clues to what could be more or less of interest for different insect species.

Answer: Accepted, the sentence was rewritten as follows: “All coffee plots were grown in places exposed to the sun, except at the Pueblo Bello Experimental Station, where coffee was grown in the shade. In Colombia coffee plots are surrounded by areas of natural vegetation, patches of secondary forest and other crops (Example Figure 2)”.

L 169 issues with font size.

Answer: Accepted

L 182, 183, 219, 220, 330, 334, 343 (and others) check that Latin names are in italics.

Answer: Accepted

It would be nice to provide a second appendix with the complete list of recorded insects in the present study. (Maybe including data such as full name, family, number of individuals, number of sites, season, and daytime?)

Answer: Accepted, was attached with the list of Flower-visiting insects in coffee crops and study sites (Appendix A1).

Providing data about pollen observation is needed, especially as you provide no figures about it.

Answer: Accepted, Figure 6 was added with the pollen load data

L 516 "all data are contained within the article or the Appendix A1." That is not yet true. So please provide all the data, as a complete list of species recorded, pollen observation, etc…

Answer: clarification, Appendix A1 now A2, refers to other flower-visiting insects in coffee reported by any study in the literature. Appendix A1 was attached with the list of Flower-visiting insects in coffee crops and study sites

References check that Latin names are in italics.

Answer: Accepted